# Use of Digital Technologies for Intensifying Knowledge Sharing

**Katarína Stachová [1], Zdenko Stacho [1,\*] , Dagmar Cagáňová [2] and Augustín Stareček [2]**

[1] Institut of Civil Society, University of SS. Cyril and Methodius in Trnava, Bučianska 4/A,
917 01 Trnava, Slovakia; katarina.stachova@ucm.sk

[2] Institute of Industrial Engineering and Management, Faculty of Materials Science and Technology in Trnava,
Slovak University of Technology in Bratislava, J. Bottu 25, 917 01 Trnava, Slovakia;
dagmar.caganova@stuba.sk (D.C.); augustin.starecek@stuba.sk (A.S.)

\* Correspondence: zdenko.stacho@ucm.sk; Tel.: +421-907-082-448

**Abstract:** The operation of companies in the current environment, introducing the concept of Industry 4.0 and the establishment and expansion of inter-company networks, as well as open and knowledge-based systems in organizations, is a precondition for success in the efficient acquisition, processing, storage, and sharing of key information. The instruments created for this purpose came to the market gradually during the Third Industrial Revolution; however, with the outbreak of the Fourth Industrial Revolution, their development underwent a leap, whereas this stage is associated with the introduction and use of big data in human resources management. The aim of this paper is to analyze the current state of organizations operating with a focus on Slovakia, as well as the importance and knowledge value of the organizations, their way of internal sharing, and the level of knowledge database use for this purpose in the context of the region in which the organizations operate. On one hand, the results show a lack of businesses orientation with regard to this issue, especially in comparison with neighboring countries; however, on the other hand, they also show relatively significant progress in this area, which may increase competitiveness internationally in the near future.

**Keywords:** digitization; knowledge sharing; human resources management; competitiveness; Industry 4.0

## 1. Introduction

The current business environment determined by globalization and the advent of the Industry 4.0 concept [1–5], where key changes are mainly related to systems, inter-machine relationships, and machine–people relationships [6–10], requires a high degree of focus on the development of disposable human resources and their competences [11–13] in an effort to ensure competitiveness.

Industry 4.0 systems are primarily based on new supporting technologies such as adaptive robotics [14], cybernetic physical systems (CPS) [15], cloud technologies [16], virtualization technologies, etc. [17]. The importance of the integrated enterprise concept in terms of competitiveness and sustainability is increasing in connection with the implementation of the Industry 4.0 concept and the emergence of both inter-company networks and open and knowledge-based systems in organizations [18–21], where the effective acquisition, processing, storage, and sharing of key information is a predictor of success. The creation of such a knowledge database is then the basis for a quick search and obtaining of variables. However, we must be careful when entering the data, and confidentiality must be taken into account before any addition to the database.

At the time of digitization, the potential of big data analysis [22–26] is a topic of expert discussions of the human resources community, because organizations have a wealth of information at their disposal, including employee demographic data, recruitment data, key performance indicator (KPI) metrics, etc. Based on a knowledge database, human resource professionals can make more objective recruitment decisions, reduce adverse effects from employee performance, support employees with a higher likelihood of loyalty to the company, or effectively train employees in line with current needs.

In an effort to meet the expectations of customers and employees, businesses are deploying intelligent features into all internal systems to create the necessary flexibility and capacity. Simple and monotonous processes are automated, while other processes become more complex and interrelated. As a result, the requirements for the competence profile of available human capital are changing. In connection with the advent of Industry 4.0, changes in the requirements for both professional and personal characteristics necessary to perform work tasks can, thus, be expected [27]. Surveys carried out to this effect indicated a priority focus on areas for *personal competencies* (ability to act autonomously on specific incentives), *social/interpersonal competences* (ability to communicate and collaborate with other employees and groups), *competencies necessary for immediate realization of ideas* (ability to translate ideas into practice), *professional competencies* (ability to find and use specific knowledge at work, process understanding, information technology (IT) security understanding) [28], *methodological competencies* (e.g., creativity, problem solving, analytical thinking), and *personality competences* (e.g., flexibility, motivation to learn, ability to work under pressure) [29]. The concept of so-called learning organization and the associated effective knowledge sharing seem to be an ideal approach to acquiring key competences of disposable human resources, while digitization itself and digital systems and applications provide optimal tools that can be used for this purpose.

The solution for remote storage and data availability is provided by the cloud computing system [30]. The role of knowledge workers in the use of IT tools in an effort to remove the boundaries between firmly defined organizational units for the benefit of the overall information flow is becoming increasingly important. In this context, digitization has a significant impact in alternative work regimes, especially when working from home with connections to business systems. There are needed programs (so-called BYOD—bring your own device), which involve creating unified electronic platforms for information sharing for distance employees, which are associated with increased demands for the security of internal data and systems [12]. Security, or more precisely IT risk, was identified by Záležáková [31] as one of the main areas of risk of Industry 4.0, highlighting the risk of security (data) costs, virus protection, protection of sensitive information and business secrets, encryption, firewall server protection, and automatic scanning.

However, data protection is not the only challenge or threat to remote data storage and availability. An important bottleneck in this context is the willingness of individuals to share knowledge and, thus, create an information base for the subsequent use of big data in practice. Organizations are now realizing the fact that knowledge sharing is important because it provides a link between individuals and organizations [32]. It is the transfer of the knowledge that individuals have that transform to the organizational level into economic and competitive value for organizations [33]. Organizations recognize the need to focus on motivating individuals to share knowledge, as these are confidential and inextricably linked to human egos and commitment and do not flow easily through the organization [34,35]. However, organizations often make the mistake of setting an environment opposite to the goal of information sharing, where employees are rewarded for owning the information. This has a demotivating effect on an individual sharing data, as being the owner of knowledge has the consequence of gaining a reward, a shift in the organizational structure, job security, etc. [34]. If individuals realize that their strength in the organization derives from the knowledge they own, it is likely that they will accumulate knowledge instead of sharing knowledge [34,36,37]. According to Brown and Woodland [38], positive results in the context of willingness to share knowledge were reported in several surveys, pointing to the need for awareness of mutuality and reciprocity. Reciprocity or knowledge transfer can facilitate knowledge sharing if individuals see that their added

value depends on the extent to which they share their own knowledge with others [33,39]. Reciprocity as a knowledge-sharing motivator means that individuals must be able to predict that knowledge-sharing will prove useful [40], although they are not exactly sure what the outcome will be [41]. The correlation between reciprocity and sharing of knowledge boundaries implies that receiving knowledge from others stimulates the flow of knowledge [40]. The relationship between knowledge sharing and incentives was further supported by studies [36,42], which found that there were significant changes in the stimulus system that encouraged individuals to share their knowledge, especially through technology. The forthcoming Fourth Industrial Revolution provides comprehensive systems and tools aimed at promoting information sharing at the individual, team, and organization levels, as well as at a cluster level [12,16,17,20,22,30]. For this reason, this paper discusses the current state of focus of organizations operating in Slovakia with regard to the importance and value of knowledge in the organization, their way of internal sharing, and the level of use of knowledge databases for this purpose.

The increase in disparities in individual regions of Slovakia represents an important barrier in the effort to build a competitive economy. Considering the increase in disparities found in other surveys in the analysis of the impact of employee involvement in innovation processes [11], awareness of the smart city concept [43], or employee motivation in creative behavior [44], in the context of individual regions, the authors of this paper found out whether similar results were published in other researches.

Other authors focusing on the issues of organizations operating in Slovakia in terms of the analysis of the principle of individualization in human resource management [12], employee motivation [37], human capital building and development [45], or the use of effective strategic tools for human resource development [46,47] confirmed increase in disparities. Based on the fact above, we analyzed individual research questions in the context of relation to the region in which the organizations operate.

## 2. Literature Review

Veber [48] defined knowledge as dynamic systems including interactions among experience, skills, facts, relationships, values, thinking processes, and importance. These systems are created by information together with experience, skills, intuition, personal imaginations, and mental models. Knowledge has two dimensions, explicit and tacit. According to Mládeková [49], explicit knowledge is expressed by formal and systematic language, via data. It is possible to verbalize, to write, or to draw it, while it can also be stored and transferred. The explicit dimension of knowledge is information. Tacit knowledge is created via the interaction of explicit knowledge with experience, skill, intuition, imaginations, mental models, etc. It is connected to activities, procedures, routines, ideas, values, or emotions of a certain person. Therefore, it is very difficult to express and share it. Its character is highly personal. The employee who is its proprietor does not have to know about it.

In essence, knowledge continuity management is based on communication. It is important for employees to know what it means to the organization when they know what the others need to know and what information should be shared; then, as a result of that knowledge, they transfer knowledge to others. While the idea itself is simple, setting up effective knowledge continuity management is a complex matter that requires technical, organizational, and managerial steps, as well as strong support from the top management. Management must understand the effort dedicated to knowledge transfer as an integral and common part of an organization's operations [50].

The employees who have the knowledge and experience should be considered as the experts in the organization, and company management should be aware of these employees. The organization should encourage its employees to transfer knowledge and experience and try to eliminate the unwillingness to share knowledge for fear of being replaced [51].

There is often knowledge loss in organizations; for example, when workers leave, there is a brain drain, which is an adverse effect. The preservation of knowledge can be achieved using strategic knowledge management and ensuring knowledge continuity. An organization's knowledge strategy determines whether the organization operates more with tacit or explicit knowledge. In predominant

work with explicit knowledge, a codification strategy is used according to References [52,53]. At present, in the era of digitization intensification, organizations have a wide range of software or applications available that facilitate the knowledge-sharing process. Among examples of technical support, it is possible to include knowledge systems, knowledge databases, e-learning applications, and electronic data interchange (EDI) support systems [54], which allow archiving and increasing the knowledge of employees provided.

Within the predominant work with tacit knowledge, a personalization strategy, creativity of employees, individual approach to the product and the customer, support for knowledge sharing, and databases have a supporting role. Hansen [52] further added that organizations applying codification strategies achieve savings in labor and communication costs through the re-use of knowledge. It should be added that References [52,53] agree that effective organizations must focus on one of these strategies and use the other in a supporting role. It is not possible to use both approaches at the same level or to reject one of these approaches completely. While ensuring the continuity of knowledge, the personalization strategy prevails.

Based on the above-mentioned findings, we focused our research attention on identifying the interest of companies operating in Slovakia in knowledge sharing, as we consider this interest to be a basic assumption for effective knowledge continuity management. On the other hand, we focused on the level of knowledge database use in order to identify the readiness of enterprises to implement either personalization or codification knowledge strategies.

## 3. Materials and Methods

We used several research and statistical methods to achieve the goal of this paper. We primarily used the analysis of current scientific research publications and confronted individual expert opinions on the issue with their own conclusions.

In order to obtain and process research data, we mainly used questionnaire surveys conducted in 2014–2018. The respondents of the survey were employees primarily responsible for human resources management and development in companies operating in Slovakia. A questionnaire with 90 questions related to human resources management was used as a tool to examine the current state of implementation of digital technologies in order to increase knowledge sharing in the enterprise. However, for the purpose of this paper, we only processed answers to questions related to knowledge sharing in the company within the context of new IT tools used. The number of respondents oscillated around 750 each year, with the rate of correctly completed questionnaires ranging from 65% to 75%. Before the actual phase of data acquisition, we set two stratification criteria based on which we subsequently specified a basic set of potential respondents. The first criterion was the region of the enterprise operation within Slovakia under the Nomenclature of Territorial Units for Statistics (NUTS) system. Specifically, Slovakia was divided into the NUTS 2 category (Bratislava region, western Slovakia, central Slovakia, eastern Slovakia), whereas the subsequent structural composition of the research sample was based on data from the Statistical Office (ŠÚ) of the Slovak Republic (SR). The second stratification criterion was the number of employees in the enterprises under investigation. The lower limit was set at 50 employees. The main purpose of establishing this stratification criterion was to primarily address to medium-sized enterprises, where we assumed, on the one hand, the existence of a formalized department and agenda for the area of human resources management and, on the other hand, given the number of employees, we predicted an increase in problems in the area of effective knowledge sharing for all employees. Based on data from the Statistical Office (ŠÚ) of SR for the period under review, it can be stated that the number of enterprises with 50 or more employees in individual regions oscillated around the same value. The specific regional structure of enterprises with more than 50 employees in the observed years is shown in Table 1.

**Table 1.** Regional structure of enterprises with over 50 employees. NUTS—Nomenclature of Territorial Units for Statistics.

| Region (NUTS II) | Bratislava Region | Western Slovakia | Central Slovakia | Eastern Slovakia |
|---|---|---|---|---|
| Regions | BA | TT, TN, NR | BB, ZA | KE, PO |
| Number of enterprises in 2013 | 1074 | 895 | 639 | 603 |
| Number of enterprises in 2014 | 1098 | 904 | 644 | 612 |
| Number of enterprises in 2015 | 1105 | 916 | 651 | 613 |
| Number of enterprises in 2016 | 1114 | 923 | 649 | 621 |
| Number of enterprises in 2017 | 1123 | 926 | 654 | 623 |

Source: own elaboration based on the Statistical Office (ŠÚ) of the Slovak Republic (SR) [55]. Note: BA—Bratislava Region, TT—Trnava region, TN—Trenčín Region, NR—Nitra Region, BB—Banská Bystrica Region, ZA—Žilina Region, KE—Košice Region, PO—Prešov Region.

The determination of the optimal research sample size from the population set out in Table 1 was performed at the 95% confidence level and confidence interval H = ±0.10. The size structure of a sufficient research sample, as well as the real size of the research sample in the analyzed years for individual regions of Slovakia, is shown in the Table 2.

**Table 2.** Determination of research sample for individual Slovak regions.

| Region (NUTS II) | Bratislava Region | West Slovakia | Central Slovakia | East Slovakia |
|---|---|---|---|---|
| Regions | BA | TT, TN, NR | BB, ZA | KE, PO |
| Sufficient size of research sample | 88 | 87 | 84 | 83 |
| Real size of research sample 2013–2017 (*n*) | 151 | 132 | 126 | 123 |

Source: own elaboration.

Companies from all sectors of economy were represented in the research each year (Table 3), whereas the research results did not show any significant differences at cross-sector comparison. For this reason, the research results were evaluated cumulatively, i.e., regardless of the sectors that companies operate in.

**Table 3.** Percentage share of companies operating in individual sectors.

| Sector | Share of Companies as a % | | | | |
|---|---|---|---|---|---|
| | 2013 | 2014 | 2015 | 2016 | 2017 |
| Manufacturing | 40.2 | 37.9 | 40.1 | 37.6 | 39.1 |
| Agriculture, forestry, and fishery | 8.2 | 9.0 | 7.3 | 10.1 | 8.0 |
| Power industry and water management | 4.4 | 3.4 | 4.1 | 2.8 | 4.2 |
| Services | 34.0 | 36.7 | 37.2 | 38.1 | 38.3 |
| Other | 13.2 | 14.0 | 11.3 | 11.4 | 10.4 |

Source: own research.

In order to evaluate the obtained data, we firstly used the methods of descriptive statistics, where we found the percentage of individual analyzed indicators. Subsequently, we statistically processed and evaluated the observed values through the analysis of basic indices (changes in values from the first year) and chain indices (changes in values monitored from the previous year) in order to identify the development trends of individual indicators over time. Finally, we verified the established hypotheses by means of a chi-square comparative analysis.

In accordance with the objectives of the paper, we formulated two research questions, on the basis of which we set up two hypotheses.

- RQ1: To what extent is knowledge shared in organizations operating in Slovakia?
- RQ2: To what extent are knowledge databases used in the conditions of Slovak organizations?
- H1: There is a statistically significant difference in the level of knowledge sharing in Slovak enterprises based on the geographical location of the organization.

- H2: There is a significant difference in the use of knowledge databases based on the geographical operation of the organization.

## 4. Results

In order to answer the research questions, we focused primarily on identifying the level of knowledge sharing in companies operating in Slovakia in the period 2013 to 2017, and we found that, in most companies, knowledge is shared, but more than half of the respondents stated that they only shared knowledge necessary to work (Table 4).

**Table 4.** Level of knowledge sharing in enterprises.

| Level of Knowledge Sharing in Enterprises | Share of Enterprises as a % | | | | |
|:---|:---:|:---:|:---:|:---:|:---:|
| | **2013** | **2014** | **2015** | **2016** | **2017** |
| Completely shared | 7.2 | 14.4 | 14.1 | 15.9 | 17.7 |
| Almost completely shared | 31.8 | 33.3 | 33.1 | 31.1 | 33.0 |
| Only the knowledge that is essential to work is shared | 59.8 | 51.6 | 50.4 | 50.5 | 49.8 |
| Not shared, knowledge is a means of securing power | 0.2 | 1.4 | 1.4 | 1.7 | 2.4 |
| Not shared, knowledge is a means of securing a monopoly from fear of losing a job | 1.0 | 0.3 | 1.0 | 0.8 | 1.1 |

Source: own elaboration.

The above findings are positive, on one hand, mainly because most respondents gave a positive response; however, on the other hand, the fact that about half of the respondents were positive only in terms of sharing knowledge needed to work is very disturbing. This result indicates a lack of readiness in companies operating within Slovakia to use disposable human resources and their potential to the maximum extent possible, which ultimately reduces their overall competitiveness vis-à-vis companies that use this potential more effectively.

We also focused on the analysis dealing with the basic index of changes in the monitored period in the area of knowledge sharing in companies (Table 5).

**Table 5.** Basic index of changes—rate of knowledge sharing in enterprises.

| Basic Index of Changes—Rate of Knowledge Sharing in Enterprises | bi14/13 | bi15/13 | bi16/13 | bi17/13 |
|:---|:---:|:---:|:---:|:---:|
| Completely shared | 2.000 | 1.958 | 2.208 | 2.458 |
| Almost completely shared | 1.047 | 1.041 | 0.984 | 1.038 |
| Only the knowledge that is essential to work is shared | 0.868 | 0.842 | 0.845 | 0.833 |
| Not shared, knowledge is a means of securing power | 7.000 | 7.000 | 8.500 | 12.000 |
| Not shared, knowledge is a means of securing a monopoly from fear of losing a job | 0.300 | 0.000 | 0.800 | 1.100 |

Source: own elaboration.

Based on the above results, we observed two trends. On one hand, there is a positive trend in the increasing number of enterprises in which knowledge among employees is fully shared at the expense of a decreasing number of enterprises in which only the knowledge needed to work is shared. On the other hand, there was a negative trend, especially in the area of information use retention with the intention to keep working in a decision-making position.

Most survey respondents in the question aimed at identifying assessment rates and forms of knowledge remuneration stated that the amount and quality of information sharing is not monitored or remunerated in any way (Table 6).

**Table 6.** Form of evaluation and remuneration of knowledge.

| In What Form Is Knowledge Evaluated and Remunerated? | Share of Enterprises as a % | | | | |
| --- | --- | --- | --- | --- | --- |
| | 2013 | 2014 | 2015 | 2016 | 2017 |
| Knowledge sharing is not tracked or remunerated | 62.6 | 63.7 | 61.2 | 62.4 | 58.7 |
| The amount and quality of shared knowledge for which the employee is remunerated in cash is closely monitored | 28.2 | 25.0 | 25.6 | 27.4 | 28.8 |
| The amount and quality of shared knowledge for which the employee is remunerated in non-cash is closely monitored | 8.1 | 9.1 | 10.9 | 8.2 | 10.6 |
| Otherwise | 1.1 | 2.2 | 2.3 | 2.0 | 1.9 |

Source: own elaboration.

The above results show a very low level of interest and initiatives of companies operating in Slovakia in the area of employee motivation to share knowledge, as only about 40% of respondents paid attention to some extent to this issue in the monitored period.

We also placed emphasis on the analysis dealing with the basic index of changes in the period under review in the field of assessment rate and forms of knowledge remuneration (Table 7).

**Table 7.** Basic index of changes—form of knowledge evaluation and remuneration.

| Basic Index of Changes—Form of Knowledge Evaluation and Remuneration | bi14/13 | bi15/13 | bi16/13 | bi17/13 |
| --- | --- | --- | --- | --- |
| Knowledge sharing is not remunerated | 1.018 | 0.977 | 0.997 | 0.938 |
| The amount and quality of shared knowledge for which the employee is remunerated in cash is closely monitored | 0.887 | 0.908 | 0.972 | 1.021 |
| The amount and quality of shared knowledge for which the employee is remunerated in non-cash is closely monitored | 1.123 | 1.346 | 1.012 | 1.309 |
| Otherwise | 2.000 | 2.091 | 1.181 | 1.727 |

Source: own elaboration.

We observed only a minimal positive increase in all the monitored attributes and, therefore, we observed that their state remained almost unchanged during the survey period.

In order to identify the degree of IT technology use in knowledge sharing, we focused our attention in the survey on the analysis dealing with the basic index of changes in the monitored period in the field of the use of knowledge databases, with respect to information systems for collecting and sharing knowledge in the enterprise (Table 8). As knowledge databases represent only a tool to support knowledge continuity management, whereas the level of their use depends on specific parameters such as the amount and form of information shared or the knowledge sharing strategy, we simplified the respondents' responses to yes/no.

**Table 8.** Basic index of changes—use of knowledge databases (information systems for collecting and sharing knowledge in an organization.

| Use of Knowledge Databases (Information Systems for Collecting and Sharing Knowledge in an Organization) | | | | |
| --- | --- | --- | --- | --- |
| 2013 | 2014 | 2015 | 2016 | 2017 |
| 22.1 | 22.7 | 24.5 | 25.9 | 31.3 |
| **Basic index of changes—use of knowledge databases (information systems for collecting and sharing knowledge in an organization** | | | | |
| bi14/13 | bi15/13 | | bi16/13 | bi17/13 |
| 1.026 | 1.111 | | 1.179 | 1.427 |

Source: own elaboration.

We observed a positive continuous increase in the monitored attribute and, therefore, we can state that its condition improved during the period of the survey; we expect a similar trend in the upcoming period.

In addition to the overall current status of the analyzed attributes or their development in the period under review in companies operating in Slovakia, we investigated whether there is a relationship between knowledge sharing and the region of business activity, where disparities in various other areas of business management were relatively significant [56,57].

- H1: Based on the chi-square test (chi = 5.826, degrees of freedom (df) = 3, $p$ = 0.12) (dichotomic variable = knowledge is not shared in the enterprise; chi categorical variable = regions), we can conclude that there is no statistically significant relationship between the variables. Based on this result, we rejected hypothesis H1 (existence of a statistically significant difference in the level of knowledge sharing in companies operating in Slovakia based on the geographical seat of the organization.

Accordingly, the statistical evaluation of dependencies under the H1 hypothesis showed that the rate of knowledge sharing is not related to the regional activity of the company. This is a positive fact, since it is possible to find a consistent approach across the country for businesses in this area. It can, therefore, be assumed that, in the case of the intensification of knowledge sharing, both within and between business entities, it can be assumed that such intensification will be equally reflected regardless of the region of activity.

- H2: Based on the chi-square test (chi = 10.675, df = 3, $p$ = 0.014) (dichotomic variable = the enterprise does/does not shared knowledge databases; chi categorical variable = regions), we can state that there is a significant relationship. Based on this result, we confirmed hypothesis H2 (existence of a significant difference in the use of knowledge databases based on the geographical activity of the organization).

Accordingly, the statistical evaluation of dependencies under the H2 hypothesis showed that the rate of use of shared knowledge databases is related to the regional activity of the company. When comparing the Bratislava region with the East Slovak region, this difference was as high as25%. These results indicate a relatively significant difference in the approach of enterprises operating in Slovakia to the implementation of new technologies in order to increase knowledge sharing across the enterprise. As a result of such an approach, disparities between businesses operating in individual regions may increase in the future, thus again affecting the level of development and economic productivity of these regions. As a result of such an approach, in the future, there may be an increase in disparities between companies operating in individual regions, which will again negatively affect the level of development and economic productivity of individual regions of Slovakia, as less developed regions (Eastern Slovakia) progress more slowly than those more developed (Bratislava region).

## 5. Discussion

The Fourth Industrial Revolution is characterized by the onset of radical changes associated with the advent of new technologies that enable the dissemination and sharing of innovation much faster than ever before [58,59]. At a time when the need for efficient data sharing is on the rise, the goal is to achieve a mostly potential positive setting on Tidd's five-tier scale of stages of building an innovative enterprise with high employee involvement. Reaching the top, this scale will ensure that every employee is fully involved in experimenting and improving things, in sharing information, and in creating an active learning organization [60]. Thus, the innovations that currently strongly determine the competitiveness of businesses are stimulated by free communication, decentralization, and trust between different hierarchical levels and, thus, the ability to communicate ideas and share knowledge across all business structures. Unfortunately, a survey based on an interview with the executives of organizations operating in Slovakia revealed that the term knowledge sharing is understood by most employees to only transmit explicit information, while this topic is not addressed in depth from the perspective of transferring practical or tacit knowledge [61]. The survey showed a low level of interest and initiatives of companies operating in Slovakia in the field of motivating employees to

share knowledge (only about 40% of participating respondents), which, in the context of the Bencsik survey [61], is negative for companies, as it can be assumed that if companies do not focus on supporting and motivating employees to share knowledge, they also do not focus on monitoring the content of shared knowledge.

The survey of this paper showed that, in 2013–2017, a positive trend of full information sharing began in enterprises, whereby the percentage of positive respondents increased from 7% to almost 18% (Table 3), which suggests a positive trend in the future. This may be reflected in the future in increasing the country's competitiveness, as the degree of knowledge sharing is also one of the factors influencing the ability to succeed. This trend is also confirmed by the positive progress of Slovakia in the Global Competitiveness Index, which analyzes the pillars of the institution, infrastructure, macroeconomic environment, health and basic education, higher education and training, commodity market efficiency, labor market efficiency, financial market maturity, technological readiness, market size, business sophistication, and innovation [62]. Based on the World Economic Forum 2017, the Slovak Republic ranked 65 out of 138 countries evaluated (Table 9) [63].

**Table 9.** Global Competitiveness Index [63].

| Global Competitiveness Index | 2012–2013 | 2015–2016 | 2016–2017 |
|---|---|---|---|
| Slovakia ranked | 71/144 | 67/140 | 65/138 |

Since the aim of the Global Competitiveness Index is to highlight what is important for the long-term growth of a country's competitiveness, the World Economic Forum reviewed the content of analyzed pillars in 2018 (institutions, infrastructure, information and communication technology (ICT) adoption, macroeconomic stability, health, skills, commodity market efficiency, labor market efficiency, financial market maturity, market size, business dynamics, ability to innovate), where Slovakia ranked 42nd in 2019 [64]. A significant shortage compared to the other analyzed countries in Slovakia is, for example, in the pillar of skills, namely, in the possibility of getting a qualified employee on the labor market (up to 127th place) [64]. This is also confirmed by the KPMG study, where Slovakia received the lowest human development index of only 3.88 out of 10, a significantly below average score (all other V4 countries gained 6.6 points or more). This implies that companies often have to settle for less qualified employees who need additional internal or external training [65]. In the context of the above, there is a need to store and share practical and tacit knowledge, which is an important variable, especially in the training of new employees during adaptation. However, the presented survey pointed out shortcomings in the current state of knowledge sharing in the analyzed organizations. On the other hand, when analyzing the level of digital skills, Slovakia is ranked 48th [64]. It follows that companies operating in Slovakia have the potential to use the incoming opportunities of the Industrial Revolution 4.0 in order to acquire and preserve skills (knowledge and information) in the company to be able to use them in the development of internal and newly acquired external employees, whereas they are currently unable to rely on the recruitment of sufficiently qualified staff from external sources. The survey presented herein pointed to significant differences in the use of knowledge databases and the geographical operation of the organization, thus confirming the existence of disparities in Slovakia, as Slovakia is one of the Organization for Economic Co-operation and Development (OECD) countries with the largest regional differences in income and unemployment. Significant diversity within the regions of Slovakia also exists in economic growth and gross domestic product (GDP) per capita, as well as in the educational level [66]. It follows from the above that it is desirable to focus on reducing the disparities of Slovakia's regions to support the use of knowledge databases in companies operating outside the Bratislava region, as this is one of the important variables in increasing the company's competitiveness, and a high percentage of employees have digital skills [64].

Several surveys conducted in Slovakia showed that businesses most often use information technologies such as the internet environment, e-mail, databases, and intranet to disseminate knowledge [67]. The above-mentioned results align with our research, which also showed an

increase in the use of knowledge databases or, more precisely, information systems for collecting and sharing knowledge in the company in 2013–2017 from 22% to more than 30% (Table 7). For the effective functioning of knowledge management, it is important to harmonize its elements into one unit. Therefore, it is necessary to create a reliable common technology infrastructure that will allow easy capture of knowledge validation, associated with the creation of a knowledge base and the subsequent effective sharing and extraction of knowledge [68]. For this purpose, companies are using or will have to start using big data processing tools. It is the use of big data processing tools that enhances the measurability of soft indicators needed for the analysis of human resources management and development. There are currently several big data analysis concepts on the market, such as Data Intelligence, Python, and others. These are predictive analytics tools that companies use to shape employee experience in relation to key business performance indicators such as customer satisfaction (net promoter score evaluated by customers), employee satisfaction (net promoter score evaluated by employees), financial performance indicators regarding share on the market [69], the probability of employees leaving, and others.

The new reality of competition will fundamentally require new ways of thinking about procedures in human resources management, changes in human resources, and changes in the mindset of people management specialists [70]. Based on expert estimates, four significant changes can be expected in the area of human resources [71]. These relate to the reification of human resources, which implies that tasks in the area of human resources management will be increasingly transferred to intelligent smart things, such as intelligent industrial manufacturing tools that can perform time and motion studies and autonomously collect and distribute data associated with normal hours and breaks, but which can also autonomously introduce employees to their functions and train them in proper use [44]. Furthermore, it is a human resources approach that uses sensors in intelligent matters to identify various data relevant to human resource management, eliminating manual labor-intensive, expensive, and error-prone data recording used so far. Changes will also occur in the human resources management datafication, given that human resources management databases will grow exponentially and the data will describe numerous human resources issues in a comprehensive and detailed way [71]. The fourth change that technological developments in human resources management will bring is the technical integration of human resources management, which will coordinate human resources management measures with operational business requirements, which will be implemented automatically [44]. The vast majority of experts expect the direct technical integration of human resources (HR) management, i.e., the direct interaction of HR software with sensors and actuators in intelligent things used by employees, e.g., to directly provide adequate training for the situation, when the demand for training occurs [71].

## 6. Conclusions

The human resources department, which was previously based on soft information and tools, is gaining a whole new perspective with the advent of the Fourth Industrial Revolution. Currently, these departments, based on available and emerging new technologies, will have a wealth of information including employee demographics, recruitment and KPI performance data, the level of knowledge and skills at work, the quality and quantity of shared information, employee loyalty, and so on. Based on these data, human resources experts can make decisions based not only on soft, but also on hard data. With the advent of the Fourth Industrial Revolution, the human resources management area gained new tools for collecting and analyzing information, enabling data-based decision-making in all dimensions of the human resources area. The basic variable in the use of new opportunities is the involvement of employees dedicated to human resources management in organizations, as well as technical support and the competence of employees to work with new tools, to select and share appropriate information through appropriate communication channels, and to use them to streamline employees. The survey presented herein focused on individual manifestations of information digitization, to which the company responds via technological innovations toward customers and employees, such as digitization of analog and biometric data, digital interaction platforms, networking, big data analytics,

rapid analytics, predictive analytics, and the use of social networks when searching and selecting employees. In this survey, the current and expected future state was analyzed, which should predict the level of awareness of existence and, accordingly, the need to acquire the competence of employees working in human resources management departments to work with new tools that can be used for both effective storage and sharing of necessary knowledge. The 2013–2017 survey revealed an insufficient focus of organizations on both knowledge sharing and the use of knowledge databases in organizations operating in Slovakia. For this reason, we feel that there is a need to focus on a more specific analysis, which was, therefore, launched in January 2020 and data collection is currently underway. The results of this survey will be published, where we will try to define the development trend in companies.

**Author Contributions:** Conceptualization, K.S. and Z.S.; methodology, A.S.; software, A.S.; validation, A.S.; formal analysis, Z.S.; investigation, Z.S.; resources, D.C.; data curation, K.S., A.S., and Z.S.; writing—original draft preparation, K.S., A.S., and Z.S.; writing—review and editing, D.C. and Z.S.; visualization, Z.S.; supervision, D.C.; project administration, K.S. and D.C.; funding acquisition, D.C. All authors read and agreed to the published version of the manuscript.

**Funding:** This research was funded by the VEGA 2/0077/19 project titled "Work competencies in the context of Industry 4.0".

**Acknowledgments:** This research was supported by project VEGA 1/0412/19 Systems of Human Resources Management in the 4.0 Industry Era, with additional support of the project APVV-17-0656 "Transformation of Paradigm in Management of Organizations in the Context of Industry 4.0", and with support of the VEGA 2/0077/19 project titled "Work competencies in the context of Industry 4.0".

**Conflicts of Interest:** The authors declare no conflicts of interest.

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
