# Peer review of "Use of Digital Technologies for Intensifying Knowledge Sharing"

_applsci, doi:10.3390/app10124281_

Round 1

Reviewer 1 Report

This paper presents an analysis of survey data related to how like Slovakian employees share information within organizations and makes the case that based on the change in time that this area is in need of improvement, potentially through Big Data.

I believe the research design is useful in presenting comparative data and that the discussion is relevant. I recommend that the authors work to better address the point they make on page two regarding the motivations for sharing information. They do not explicate in their data and argument how Big Data will counter their stated reasons for employees not sharing data and specifically note that there are emotional and human reasons why this is the case.

Author Response

  • We accepted the comment and added the required on the lines 78-90

Reviewer 2 Report

The paper presents an interesting approach to the subject of Industry 4.0, business networks and the necessary management of key information, with bibliography and current approaches. Furthermore, it can be very useful for the country, where the research is carried out, to know its current position and, consequently, to take future actions and improve its competitiveness.

However, there are some aspects that I consider should be improved to be able to publish in a journal of this level:

  • The paper, except in the introduction, does not present a section on the state of the art, which is essential in order to carry out a correct discussion and obtain coherent conclusions. A further development of the previous literature regarding the analyzed aspects would be necessary.
  • The difference between the concepts of Knowledge sharing and Knowledge database is not clear. It is essential that these concepts are very clear from the beginning to understand the hypotheses analyzed.
  • Based on the above, it is not understood how Hypothesis 1 is not fulfilled and the author takes this as something positive, in the sense that there is no relationship with the region where the company is based, so there will be no differences between the development of the different areas of the country. However, Hypothesis 2 is fulfilled but it is argued that this is also positive and that it will allow the development of these areas. I see a contradiction, in the sense that it affirms that two contrary things are both positive. All this is not at all clear.
  • A clear relationship between the Global Competitiveness Index and the sharing of knowledge is taken for granted, but I consider that one thing does not have to completely justify the other because there may be many other variables that increase competitiveness and that have not been analyzed in this article. It would be necessary to mention this fact and comment on the existence of other variables.
  • This is a very simple empirical study (possibly because it is a fragment of a larger one) but what is offered here are few questions. In addition the scale used in the answers can confuse the interviewee, because the borders between one and the other answers are not entirely clear. And, in other cases, however, the answer is yes or no, without giving possibility to intermediate situations.
  • There is a kind of self-citation from the authors, but it does not seem that it offers any use. It would be interesting to describe a little those previous works that the authors of this article have carried out or, on the contrary, not to be cited: “Considering the disparities in other surveys in Slovakia, the authors of 98 the paper [11, 43, 44] and other authors focusing on the issues of organizations operating in Slovakia 99 [12, 37, 45 -47], confirmed, we analyze individual research questions also in the context of relation to 100 the region in which the organizations opérate”
  • There are certain conclusions that do not seem to be obtained from the development of the work: e.g. “digitization of analog and biometric data, digital interaction platforms, networking, big data analytics…, rapid analytics, predictive analytics, the use of social networks when searching and selecting employees”. In addition, the discussion does not make a good treatment of the results obtained, not relating them to the previous theoretical description. Both sections should be profoundly reformulated

Author Response

The difference between the concepts of Knowledge sharing and Knowledge database is not clear. It is essential that these concepts are very clear from the beginning to understand the hypotheses analyzed.

We accepted the comment and added the required on the lines 41 – 50, 78-90

Based on the above, it is not understood how Hypothesis 1 is not fulfilled and the author takes this as something positive, in the sense that there is no relationship with the region where the company is based, so there will be no differences between the development of the different areas of the country. However, Hypothesis 2 is fulfilled but it is argued that this is also positive and that it will allow the development of these areas. I see a contradiction, in the sense that it affirms that two contrary things are both positive. All this is not at all clear.

We accepted the comment and added the required on the lines 249 - 253

A clear relationship between the Global Competitiveness Index and the sharing of knowledge is taken for granted, but I consider that one thing does not have to completely justify the other because there may be many other variables that increase competitiveness and that have not been analyzed in this article. It would be necessary to mention this fact and comment on the existence of other variables.

We accepted the comment and added the required on the lines 268-278

This is a very simple empirical study (possibly because it is a fragment of a larger one) but what is offered here are few questions. In addition the scale used in the answers can confuse the interviewee, because the borders between one and the other answers are not entirely clear. And, in other cases, however, the answer is yes or no, without giving possibility to intermediate situations.

We agree with that, but nothing can be done about

There is a kind of self-citation from the authors, but it does not seem that it offers any use. It would be interesting to describe a little those previous works that the authors of this article have carried out or, on the contrary, not to be cited

We accepted the comment and added the required on the lines 107 - 114

There are certain conclusions that do not seem to be obtained from the development of the work: e.g. “digitization of analog and biometric data, digital interaction platforms, networking, big data analytics…, rapid analytics, predictive analytics, the use of social networks when searching and selecting employees”.

We accepted the comment and added the required on the lines 359-376

the discussion does not make a good treatment of the results obtained, not relating them to the previous theoretical description. Both sections should be profoundly reformulated

We accepted the comment and added the required

Reviewer 3 Report

This paper describes a research about the use of digital technologies to intensify knowledge sharing in companies with a special focus on Slovak enterprises.

Major comments:

  • own elaboration based on ŠÚ SR – what is “ŠÚ SR” and insert a reference to the source
  • there is no information about the sample itself to understand if all companies are operating in similar or completely different industries, which has a great impact on the interpretation of the results
  • please address in the discussion the limitations of the study and the implication for academia as well as for practitioners
  • (how) can the results be generalized also to companies outside of Slovakia?

Minor comments:

  • introduce KPI
  • the authors should introduce the structure of the paper at the end of the introduction

Author Response

own elaboration based on ŠÚ SR – what is “ŠÚ SR” and insert a reference to the source

We accepted the comment and added the required

there is no information about the sample itself to understand if all companies are operating in similar or completely different industries, which has a great impact on the interpretation of the results

We accepted the comment and added the required on the lines 151-156

please address in the discussion the limitations of the study and the implication for academia as well as for practitioners (how) can the results be generalized also to companies outside of Slovakia?

We accepted the comment and added the required on the lines 295-299

Reviewer 4 Report

The paper Use of digital technologies with the aim of  intensifying knowledge sharing in companies operating in Slovakia is up to date, the authors have defined the right goal, research questions and hypotheses. The structure of the paper is logical. The research methods are described adequatelly. 

The results are responding research.

Author Response

Thanks for rewiev

Reviewer 5 Report

I believe that most of the paper is coherent and well-written. However, there are some minor issues to solve:

  1. Title:I suggest the following one: Use of digital technologies with the aim of 3 for intensifying knowledge sharing in companies 4 operating in Slovakia
  2. abstract: is ok.
  3. Introduction. I will fit some of the following aspects:

3.1

In the line 78 there is a missing word "resp. "

3.2. line 107-115.  I believe that this paragraph is too long and it is difficult to understand the mix of ideas and objectives. Maybe divide the ideas an several paragraphs help to understand future readers.

"Considering the increase disparities in other surveys in Slovakia, the authors of the paper in disparities in other surveys in Slovakia, and in the analysis of the impact of employee involvement in innovation processes [11], awareness of the Smart City Concept [43] or employee motivation in creative behaviour [44], in the context of individual regions of and other authors focusing on the issues of organizations operating in Slovakia in the analysis of the principle of individualization in human resource management [12], employee motivation [37], human capital building and development [45] or the use of effective strategic tools for human resource development [46, 47], confirmed, we analyze individual research questions also in the context of relation to the region in which the organizations operate"

I missed some literature review to highlight the relevance about the topic between the introduction and methods. For example a theoretical framework and its evolution as well as the situation in emerging countries. Another interesting section will be the link between knowledge sharing and industry 4.0. Both sections can help to explain the gap and the relevance of the paper. After literature review I will introduce a visual figure to facilitate the understanding by readers of the hypotheses (Conceptual Model Figure).

I will explain that these codes/ acronyms BA TT, TN, NR BB, ZA KE, PO are for identifying the regions. I realised of many tables, maybe some of them cab be joined (For example 1 and 2), or 3 and 4., 6, 7 and 8. 

Regarding hypotheses I will explain them (Research Question and Hypotheses), for not overlapping them.

I see value to the explanation of results regarding hypotheses (lines 221-253).

The new introduction of red paragraphs provide a better explanation of the phenomenon in discussion and conclusions. 

Author Response

Thanks for the review:

  • we have modified the title of the article according to your proposal
  • we have tried to reformulate paragraph in lines 107 - 118 to make it clearer
  • we explained that these codes/acronyms BA TT, TN, NR BB, ZA KE, PO in the note under Table 1
  • We have added the part of article: Literature review in lines 120-164

Round 2

Reviewer 2 Report

Unresolved: The difference between the concepts of Knowledge sharing and Knowledge database is still not clear. It is essential that these concepts are very clear from the beginning to understand the hypotheses analyzed.

Unresolved: A clear relationship between the Global Competitiveness Index and the sharing of knowledge is taken for granted, but I consider that one thing does not have to completely justify the other because there may be many other (the author only speak about one of them) variables that increase competitiveness and that have not been analyzed in this article. It would be necessary to mention this fact and comment on the existence of other variables.

Unresolved (the author states the impossibility of doing something about it): This is a very simple empirical study (possibly because it is a fragment of a larger one) but what is offered here are few questions. In addition, the scale used in the answers can confuse the interviewee, because the borders between one and the other answers are not entirely clear. And, in other cases, however, the answer is yes or no, without giving possibility to intermediate situations.

Unresolved: The discussion does not make a good treatment of the results obtained, not relating them to the previous theoretical description. Both sections should be profoundly reformulated

Author Response

Thanks for review:

  • We describe the difference between the concepts of Knowledge sharing and Knowledge database in part Literature review lines 120-164
  • We accepted the comment and we mention this fact and comment on the existence of other variables in lines 331-343
  • We have added an explanation of the used scales of answers in lines265-268
  • The parts Discussion, Introduction and Literature review was reformulated

Reviewer 3 Report

Dear authors. Thank you for considering the suggestions for improvement.

Author Response

Thank you for the review.